# Effect of Plant Growth Regulators on Cotton Seedling Root Growth Parameters and Enzyme Activity

**DOI:** 10.3390/plants11212964

**Published:** 2022-11-03

**Authors:** Zhenxi Cao, Xingpeng Wang, Yang Gao

**Affiliations:** 1College of Water Conservancy and Architecture Engineering, Tarim University, Alaer 843300, China; 2Laboratory of Modern Agricultural Engineering, Tarim University, Alaer 843300, China; 3Key Laboratory of Northwest Oasis Water-Saving Agriculture, Ministry of Agriculture and Rural Affairs, Shihezi 832000, China; 4Institute of Farmland Irrigation, Chinese Academy of Agricultural Sciences, Xinxiang 453002, China

**Keywords:** seedling stage, root system, growth regulator, growth parameters, enzyme activity

## Abstract

It is well known that the survival rate of cotton seedlings is low, and the growth and development status at this stage is crucial to improve productivity. Plant hormones are important factors that promote the growth and development of cotton seedlings. Growth regulators have the same function as plant hormones. The purpose of this research is to explore the effects of different concentrations of growth regulators on cotton root morphological parameters and enzyme activities, and to find suitable plant growth regulators and their optimal concentrations to improve the growth of the cotton seedling root system. Three cotton varieties, “Zhongmian 619” (Z619), “Xinluzao 27” (Z27), and “Xinluzao 39” (Z39), and three growth regulators, gibberellin (GA_3_), salicylic acid (SA), and paclobutrazol (PP_333_), at three concentrations were used in our experiment. In Z619 and Z27, 0.050 mg/L GA_3_ significantly increased the total root length. Similarly, 0.010 mmol/L SA significantly increased the root growth parameters of Z619 and Z39. In Z619, 0.1 mg/L PP_333_ significantly increased the total root length and total surface area and reduced the average root diameter. For all three cotton varieties, 0.050 mg/L GA_3_ increased peroxidase (POD) activity in the roots. In Z27 and Z39, 0.80 mg/L GA_3_ increased superoxide dismutase (SOD) activity in the roots. All SA concentrations increased SOD activity in roots of Z619 and Z27; 0.10 mg/L PP_333_ significantly increased SOD and POD activities in the roots of Z619 and significantly increased SOD activity in Z27. Principal component analysis indicated that 0.10 mmol/L SA was the optimal treatment for promoting the development of the roots of Z619 and 0.050 mmol/L SA was the optimal treatment for promoting the development of the roots of Z27 and Z39.

## 1. Introduction

The Tarim Basin is a typical oasis irrigation area with sufficient light and heat resources and is suitable for growing cotton. While adult cotton plants are resistant to stress, cotton seedlings are vulnerable. The low survival rate of seedlings is a serious problem in cotton cultivation [1]. In this area, improvements to the planting system may effectively increase the germination rate of cotton seeds, but there is a lack of related agricultural technology to improve the seedling survival rate. At the seedling stage of cotton plants, the nutrients used to maintain seedling growth are mainly absorbed and transported by the roots [2]. Root systems are significantly affected by adverse soil environments, which may weaken the growth and development of plants and even reduce the survival rate of cotton seedlings [3]. Improving the physiological parameters of the growth of cotton root systems to enhance their ability to absorb soil water and nutrients may promote the growth and development of cotton plants throughout the growth period and increase yield [4]. Therefore, improving the physiological parameters related to the growth of the cotton seedling root system is of great significance in guiding cotton production in the Tarim Basin.

Plant growth regulators can promote plant growth and development processes, such as cell division, rhizome elongation, flower and fruit development [5,6,7,8,9]. During cotton plant growth, the dramatic change in the soil environment may cause the production of a large amount of reactive oxygen species (ROS), which can damage the cell membrane systems or cause cell death. Plants have a defense system to cope with ROS [10]. Superoxide dismutase (SOD), peroxidase (POD), and catalase (CAT) are important enzymes that protect the cell membrane system. SOD first catalyzes the decomposition of intracellular ROS O^2−^ into H_2_O_2_, which is then decomposed by intracellular CAT and POD. As a result, the destruction of the membrane system by ROS is minimized, and thus the damage and death of plant cells are reduced [11]. Growth regulators have the same function as plant hormones and can be applied in the soil to modulate the activities of related enzymes in plants to balance the metabolism of ROS, the roles of which differ in different growth stages, tissues, and environments [12,13,14]. Appropriate gibberellin (GA_3_) [15] and salicylic acid (SA) concentrations can significantly promote the elongation and growth of plant cells, while paclobutrazol (PP_333_) may restrict plant growth. Sponsel et al. [16] found that wheat seeds could effectively improve the antioxidant enzyme activity in seeds after gibberellin treatment. Martin et al. [17] showed that the yield of litchi trees treated with gibberellin could be increased without changing their quality. The alleviation of NaCl toxicity by SA was related to the activities of SOD and CAT, as well as the increase of the contents of ascorbase and glutathione [18]. The SA-treated thyme plants had greater shoot and root dry weights, photosynthetic rates, mesophyll efficiency, and water-use efficiency when exposed to salt stress [19]. Paclobutrazol can effectively improve the growth, physiological characteristics and stress resistance of crops by regulating their biomass distribution, water status, cell permeability, and oxidation resistance [20,21,22]. Choudhary et al. [23] found that soaking maize seeds in PP_333_ solution effectively reduced the height of maize seedlings, promoted root growth, and increased the root–shoot ratio and dry matter of plants. In addition, the activity of protective enzymes in leaves was significantly increased, and the stress resistance of seedlings was enhanced, which benefited the growth and development of the seedlings. PP_333_ increased the soluble protein and soluble sugar content, enhanced SOD, POD, and nitrate reductase activities, and reduced malondialdehyde (MDA) content. PP_333_ can also significantly increase peroxidase and indoleacetic acid oxidase activities in cotton roots and cotyledons, and the activity of the enzymes in taproots is higher than that in cotyledons. The effects of growth regulators on the growth of roots and the activities of key enzymes may vary with crop, species, growth regulator species, and growth regulator concentration. Previous studies have demonstrated that the application of growth regulators under salt stress conditions effectively promotes the growth of the seedling roots of crops.

A well-developed root system in cotton plants ensures a high yield. Growth regulators can promote seedling root growth. Thus far, the effect of different growth regulators on the root growth of cotton seedlings growing in the Tarim Basin has rarely been reported. Therefore, it is of great significance to study the effects of growth regulators on the growth and development of cotton seedling roots. Accordingly, the main goal of this study was to analyze the impact of GA_3_, SA and PP_333_ on cotton physiological growth indicators, explore the appropriate concentration of growth regulators to promote the growth of different cotton seedlings, and provide a reference for improving the growth of cotton seedlings in Xinjiang and increasing the yield of growth regulators.

## 2. Results

### 2.1. Effect of the Growth Regulator Concentrations on the Root Growth of the Seedlings

#### 2.1.1. Effect of the GA_3_ Concentration on the Root Growth

Figure 1 shows the effects of different GA_3_ concentrations on the root morphology of the three varieties, “Zhongmian 619” (Z619), “Xinluzao 27” (Z27), and “Xinluzao 39” (Z39). Data analysis indicated that the G1 (GA_3_ 0.050 mg/L) concentration increased the total root length of Z619 by 42.2% compared with CK (control check), and the difference reached a significant level, indicating that the G1 concentration of GA_3_ significantly promoted root elongation. G2 (GA_3_ 0.20 mg/L) and G3 (GA_3_ 0.80 mg/L) had no significant effect on the total root length. The low GA_3_ concentration had no significant effect on the total root surface area or volume. There was no significant difference in the total root surface area and volume between G1 and CK, while G2 and G3 significantly reduced the total root surface area and volume compared with CK, indicating that an increase in the GA_3_ concentration restrained the increase in the total root surface area and volume. All GA_3_ concentrations significantly reduced the average root diameter of the seedling compared with CK, indicating that GA_3_ significantly inhibited the growth of root thickness.

There was no significant difference in the total root surface area and volume of Z27 between GA_3_ concentrations. The low GA_3_ concentration significantly increased the total root length by 25.6% compared with CK. All GA_3_ concentrations significantly reduced the average root diameter compared with CK.

The low GA_3_ concentration showed no significant effect on the total length, total surface area, or volume of Z39 roots. The G1 concentration reduced the average diameter by 18.0% compared with CK, and the difference was significant, indicating that the low GA_3_ concentration may have reduced the root diameter of Z39. There was no significant difference in the morphological parameters between G2 and CK, indicating that the G2 concentration of GA_3_ had no effect on the root morphology. The high GA_3_ concentration significantly increased the volume but significantly reduced the average root diameter.

#### 2.1.2. Effect of SA Concentration on the Root Growth

Figure 2 shows the effects of different concentrations of SA on the root morphology of three varieties: Z619, Z27, and Z39. Different SA concentrations significantly affected all root morphological parameters of Z619, which included the increase in total length, total surface area, and volume, and the reduction in average diameter of the roots.

While significantly reducing the root volume and average root diameter of Z27, the low SA concentration had no significant effect on the total length or total surface area. Compared with CK, the S1 (SA 0.010 mg/L) concentration of SA increased the root volume and average root diameter by 18.1% and 21.1%, respectively. In comparison with CK, the S2 (SA 0.050 mg/L) concentration of SA increased the total length, total surface area, and volume of the roots, but the differences were not significant; however, this concentration significantly reduced the average root diameter. The high SA concentration reduced the total length, total surface area, and volume of roots but had no significant effect on the average root diameter. While significantly reducing the root volume by 16.7% in comparison with CK, the S3 (SA 0.10 mg/L) concentration of SA showed no significant effect on the total length, total surface area, and average diameter of the roots.

The low SA concentration significantly increased the total length, total surface area, and volume, but reduced the average root diameter of Z39. Compared with CK, the S1 concentration of SA significantly increased the total length, total surface area, and volume of roots by 35.2%, 20.6%, and 19.5%, respectively, and significantly reduced the average root diameter by 18.9%. While the S2 concentration of SA had no significant effect on the total root length and root volume, it significantly reduced the total root surface area and average root diameter by 18.1% and 16.2%, respectively. The high SA concentration significantly reduced the total root length and surface area but had no significant effect on the root volume or average root diameter.

#### 2.1.3. Effect of the PP_333_ Concentration on the Root Morphology

Figure 3 shows the effects of different PP_333_ concentrations on the root morphology of three varieties: Z619, Z27, and Z39. The effect of PP_333_ on the root morphological parameters varied depending on the concentration. The low PP_333_ concentration significantly increased the total root length and surface area but reduced the average root diameter of Z619. The P1 (PP_333_ 0.10 mg/L) concentration of PP_333_ increased the total root length and surface area by 16.2% and 21.2%, respectively, and reduced the average root diameter by 12.8% compared with CK. There was no significant difference in the root morphological parameters between P2 (PP_333_ 0.250 mg/L), P3 (PP_333_ 0.50 mg/L), and CK, except for the significantly reduced total root length induced by the P3 concentration of PP_333_.

All PP_333_ concentrations showed inhibitory effects on the total length, total surface area, and volume of the roots of Z27. The P1, P2, and P3 concentrations of PP_333_ significantly reduced the total length, total surface area, and volume of the roots compared with CK. In terms of the average root diameter, while it was significantly increased by the P2 concentration of PP_333_, there was no significant difference between P1, P3, and CK, showing a first increasing and then decreasing trend with the increase in PP_333_ concentration.

The P1 concentration of PP_333_ had no significant effect on the root morphological parameters of Z39. As the concentration increased, PP_333_ showed inhibitory effects on the total length, total surface area, and volume but increased the average diameter of the roots. In comparison with CK, the P2 and P3 concentrations of PP_333_ significantly reduced the total length, total surface area, and volume but significantly increased the average diameter of the roots. With the increase in concentration, the inhibitory effect of PP_333_ on the total length, total surface area, and volume of the roots increased, and the promoting effect of PP_333_ on the average root diameter became stronger.

### 2.2. Effects of Different Concentrations of Growth Regulators on Root Enzyme Activities in the Seedling Stage

#### 2.2.1. Effects of the GA_3_ Concentration on the Activities of Root-Related Enzymes

Most soluble proteins in a cotton plant are enzymes that are involved in various metabolic processes, the content of which is an important indicator of the overall metabolic level of a cotton plant. Figure 4 shows the effects of different GA_3_ concentrations on the activities of root-related enzymes of Z619, Z27, and Z39. GA_3_ had a significant effect on the soluble protein content of the roots of Z619. With an increase in the GA_3_ concentration, the soluble protein content significantly increased. The high GA_3_ concentration significantly promoted the formation of soluble proteins in the roots of Z27. Similar to the response of Z619, with the increase in GA_3_ concentration, the soluble protein content of the roots of Z39 gradually increased. Compared with CK, the G1 concentration significantly reduced the soluble protein content by 23.1%. The G2 and G3 concentrations of GA_3_ significantly increased the soluble protein content by 13.9% and 25.0%, respectively. The low GA_3_ concentration suppressed the formation of soluble proteins in the roots, but the high concentration promoted the synthesis of soluble proteins.

While the low GA_3_ concentration had no significant effect on POD activity in the roots of Z619, as the GA_3_ concentration increased there was a significant inhibitory effect on POD activity. With the increase of GA_3_ concentration, POD activity in the Z27 roots was enhanced, but the G3 concentration inhibited POD activity. Similar to the response of Z619, POD activity in the Z39 roots decreased significantly with the increase in GA_3_ concentration, and the order of POD activity at the GA_3_ concentrations was G1 > G2 > G3. The low GA_3_ concentration significantly increased POD activity in the Z39 roots. However, with the increase in the concentration, the promoting effect of GA_3_ gradually decreased and a significant inhibitory effect on POD activity appeared.

GA_3_ showed an inhibitory effect on SOD activity in Z619 roots. In contrast to the effect on Z619, the low GA_3_ concentration had no significant effect on SOD activity in Z27 roots, while G2 and G3 increased the SOD activity by 18.9% compared with CK. Compared with CK, the low (G1) and high (G3) concentrations of GA_3_ significantly increased SOD activity in the Z39 roots by 32.9% and 31.1%, respectively.

#### 2.2.2. Effect of SA Concentration on the Activity of Root-Related Enzymes

Figure 5 shows the effects of different SA concentrations on the activities of root-related enzymes in Z619, Z27, and Z39. Dissimilar to the effect of GA_3_, the S2 and S3 concentrations of SA significantly increased the soluble protein content of the Z619 roots by 22.2% and 25.0%, respectively, compared with CK. The low SA concentration significantly inhibited the formation of soluble proteins in the Z27 roots, and the increased SA concentrations had no significant effect on the soluble protein content. The response of Z39 to SA was different from that of Z619 and Z27. For Z39, while the S2 concentration of SA increased the root soluble protein content by 18.5% compared with CK, low and high concentrations of SA had no significant effects.

The effects of SA on POD activity in the Z619 roots varied with the concentration. There was no significant difference between S1 and CK. The S2 and S3 concentrations of SA significantly reduced the POD activity by 13.4% and 33.0%, respectively, compared with CK. With the increase in SA concentration, the POD activity in the Z27 roots showed a decreasing trend, and the order of POD activity between the SA concentrations was S1 > S2 > S3, indicating that the low SA concentration increased the POD activity in the Z27 roots. None of the concentrations (S1, S2, and S3) of SA showed significant effects on POD activity in the Z39 roots compared with CK.

In contrast to the effects of GA_3_, all SA concentrations increased SOD activity in the Z619 roots. SOD activity was significantly increased by 20.6%, 37.2%, and 29.4% at the S1, S2, and S3 concentrations of SA, respectively. All SA concentrations significantly increased SOD activity in the Z27 roots, and the effect was gradually weakened with the increase in SA concentration. The S1, S2, and S3 concentrations of SA increased SOD activity by 47.7%, 36.3%, and 24.1%, respectively. The S2 concentration of SA increased SOD activity in the Z39 roots by 22.8% compared with CK.

#### 2.2.3. Effect of the PP_333_ Concentration on the Activities of Root-Related Enzymes

Figure 6 shows the effects of different PP_333_ concentrations on the activities of root-related enzymes in the Z619, Z27, and Z39 roots. Similar to the effects of SA, while the high PP_333_ concentration reduced the root soluble protein content of Z619, the low concentration had no significant effect. None of the PP_333_ concentrations had a significant effect on the soluble protein content of the Z27 and Z39 roots. There was no significant difference in soluble protein content between P1, P2, P3, and CK.

Both the P1 and P3 concentrations of PP_333_ increased the POD activity in the Z619 roots. Compared with CK, the P1 and P3 concentrations of PP_333_ significantly increased the POD activity by 12.1% and 22.7%, respectively. The difference in POD activity between P2 and CK was not significant. With the increase in PP_333_ concentration, the POD activity in the Z27 roots first decreased and then increased. With the increase in PP_333_ concentration, the POD activity in the Z39 roots increased first and then decreased. The P2 concentration of PP_333_ significantly increased POD activity, while the P1 and P3 concentrations had no significant effects.

The low PP_333_ concentration promoted SOD activity in the Z619 roots. The P1 concentration of PP_333_ significantly increased SOD activity by 20.4% compared with CK. With an increase in concentration, the promoting effect of PP_333_ gradually decreased. Similar to the effect of SA, all PP_333_ concentrations increased SOD activity in the Z27 roots to a certain extent, and with the increase in the PP_333_ concentration, SOD activity showed a decreasing trend. Compared with CK, the P1, P2, and P3 concentrations of PP_333_ increased SOD activity by 70.7%, 42.0%, and 16.0%, respectively, and the increases caused by P1 and P2 were significant. All PP_333_ concentrations increased SOD activity in the Z39 roots. Compared with CK, the P1, P2, and P3 concentrations of PP_333_ increased SOD activity by 33.6%, 18.2%, and 22.9%, respectively, but the increases were not significant.

### 2.3. Single Factor ANOVA of Varieties and Growth Regulators

Single factor analysis of variance (ANOVA) of the root growth parameters was performed to clarify the influence of regulators and varieties. It was found that different varieties of cotton had no significant effect on physiological growth (*p* > 0.05). It shows that the difference of physiological growth indexes among different cotton varieties is not obvious. The indexes with obvious difference of different growth regulators on cotton physiological growth indexes are root length and average heel diameter, and their average values are shown in Figure 7. The effect of different growth regulators on cotton root length was significant (*p* < 0.001). It can be seen that there was a significant difference between SA treatment and PP_333_ treatment. According to the data, SA can promote the growth of cotton root length, and PP_333_ can inhibit the growth of root length to some extent. The effect of different growth regulators on the average diameter of cotton heel was significant (*p* < 0.01). It can be seen that there was a significant difference between GA_3_ treatment and PP_333_ treatment. According to the data, PP_333_ can promote the growth of cotton root length, and GA_3_ can inhibit the growth of cotton root length to some extent. In this study, the main indexes of root length and average root diameter were the effects of growth regulators on cotton physiological growth. These two indicators can be used as the main research indicators of growth regulators on cotton.

### 2.4. Principal Component Analysis of the Root Growth Parameters of Cotton under Treatment with Different Growth Regulators

After data standardization, the principal components were computed using SPSS 25 software. The principal component analysis diagram is shown in Figure 8. The main components of growth regulators affecting cotton physiological growth are the main components of morphological factors and protein factors. There was a certain degree of aggregation among the treatments, but the three growth regulators had obvious separation. The similarity of 27 treatments was not high, indicating that the three growth regulators had different effects on cotton physiological growth indicators. The eigenvectors and cumulative contributions of the sample correlation matrix are shown in Appendix A. In terms of growth regulators, the contributions of the first three principal components were 44.55%, 28.19%, and 15.29%, respectively, indicating that PC1, PC2, and PC3 explained 88.03% of the variation in the seven original variables. The initial data of the seven original variables were transformed and three new independent comprehensive variables (principal components) were obtained, in which the eigenvalue of root surface area among morphological factors was the largest (0.976). The eigenvalue of soluble protein content among protein factors was the largest (−0.935), and the eigenvalue of SOD among enzyme activity factors was the largest (0.956). The results indicated that root surface area, soluble protein content, and SOD activity had the highest contributions and a high degree of influence, which could be used as major indicators for studying the physiological effects of growth regulators on cotton growth.

Based on the principal component analysis of factors, the scores of the variety–growth regulator factors were ranked, as shown in Appendix A. Among the ranked morphological factors, the top three were Z39-S3, Z619-S1, and Z619-S3, respectively; among the ranked protein content factors, the top three were Z619-S3, Z619-P3, and Z39-G1; among the ranked of enzyme activity factors, the top three were Z27-P1, Z619-S2, and Z27-P2; among the ranked comprehensive score of the principal components, the top three were Z619-S3, Z39-G1, and Z619-P1. The scores listed in the table indicated that the optimal concentration of the growth regulators varied by variety; the SA score calculated based on the three varieties was higher than that of the other two growth regulators; root surface area and soluble protein content were the main morphological and protein content factors, respectively. The results could be used as a basis for selecting the growth regulator concentration to promote the growth and development of seedlings of different cotton varieties.

## 3. Discussion

Plant hormones are important factors affecting the growth and development of the root system. Plant growth regulators can influence plant stress resistance by affecting the action of endogenous plant hormones [24]. In our study, the effects of three growth regulators on the root growth parameters of cotton seedlings and the activities of root-related enzymes were determined, and the response of the seedling root system to different concentrations of regulators was examined. By combining the observed data of morphological parameters and enzyme activities with principal component analysis under the treatment of growth regulators, there was a negative correlation between the elongation and thickening of the seedling root system.

GA is considered as an effective growth regulator in some studies, which can improve crop yield and economic income [25,26]. Aging and various environmental stresses may disrupt the balance between the generation and detoxification of reactive oxygen species [27]. Achard et al. showed that GA could promote the production of DELLA proteins, which promoted the expression of SOD, thereby reducing the ROS level [28]. In our study, the low GA_3_ concentration (0.050 mg/L) level promoted the elongation of root cells and thus increased the root length; however, the growth of average root diameter was inhibited, resulting in restrained growth of the root surface area and volume, which led to poor robustness of the root system. Therefore, high GA_3_ concentration levels may cause lodging. Varieties with higher GA_3_ sensitivity might exhibit restrained root elongation at different concentrations, and excessively high GA_3_ concentrations inhibited root growth [29]. An appropriate SA concentration increased all the growth parameters of the roots of all varieties. SA can induce cell differentiation and promote plant growth and development. Appropriate SA concentrations promoted both elongation and thickening of the cotton seedling roots and did not restrain either of them. Therefore, SA is a stable plant growth regulator. However, while low concentration of SA can promote root growth, high concentration of SA can inhibit root growth. The same results were obtained with chamomile testing [30]. In this study, through two-way interaction analysis and principal component analysis, the comprehensive score of SA was found to be higher than that of the other two growth regulators; the root surface area was the most important parameter of root morphology, and the soluble protein content was the most important index of protein content. Principal component analysis [31] indicated that SA application was beneficial to the seedling roots of the three varieties in terms of growth and enzyme activity [32]. The low PP_333_ concentration (0.10 mg/L) promoted root elongation. Similar results were obtained with geranium plants [33]. However, its promotion extent was lower in comparison to other growth regulators. With the increase in the PP_333_ concentration, the root length was reduced. Additionally, PP_333_ did not have a distinct effect on the thickness of seedling roots. It is possible that PP_333_ may inhibit GA_3_ synthesis [34], thereby inhibiting root elongation. The low PP_333_ concentration had no significant inhibitory effect on root elongation, and the root length still increased if the variety was not sensitive to PP_333_. The increased PP_333_ concentration inhibited the growth of root length. Moreover, PP_333_ could remain in the plants [35]. In follow-up studies, we will observe the duration of PP_333_ retention, compare the root growth in different growth stages of cotton plants, and examine the duration of the effect of PP_333_ residue in plants on the root growth, so as to promote the increase of flowering and fruiting of crops [36,37].

The sensitivity of the root-related enzymes to growth regulators varied with the cotton variety. Different SA concentrations increased the antioxidant enzyme activity and soluble protein content in the roots to varying degrees, which could effectively increase the vitality of the cotton roots. PP_333_ had no effect on the soluble protein content of the roots of any of the varieties. The SOD activity in the roots of the three varieties was not significantly affected by GA_3_. Furthermore, compared with the activity of the other two root-related enzymes, the POD activity was less affected by the three growth regulators, indicating that the POD activity was less sensitive to growth regulators. In a certain range of concentrations, the POD activity in the roots was highly stable.

Analysis of the effects of the interactions between growth regulators and varieties on root growth parameters showed that the growth parameters were not significantly affected by the variety, indicating that the changes in root growth parameters were unrelated to variety. Growth regulators had significant effects on the root length and average root diameter. Root volume and SOD activity were significantly affected by the interaction between growth regulators and varieties. The choice of growth regulator was the major contributing factor to the root growth parameters, and their effects on the parameters varied with variety.

To improve the growth of cotton seedlings in the Tarim Basin and ensure good growth of adult plants and high yield, 0.10 mmol/L SA can be applied to seedlings of variety Z619, and 0.050 mmol/L SA can be applied to seedlings of varieties Z27 and Z39. The results of this study provide a reference for the improvement of cotton seedling growth through chemical modulation in the Tarim Basin.

## 4. Materials and Methods

### 4.1. Materials and Experimental Design

As shown in Table 1, three cotton varieties (Z619, Z27 and Z39) and 3 growth regulators (GA_3_, SA and PP_333_) were used in the experiment. Three concentrations of each growth regulator and a control without any regulator were set for each variety. A total of 30 treatments were set, and each treatment had 3 replications. The experimental treatments are shown in Table 2.

The prepared GA_3_, SA, and PP_333_ solutions were added to the culture substrates with a 35% water content. Three sterilized seeds were planted in each replicate and then covered with the substrate to 3 cm in thickness. The compactness of the substrates was adjusted to a level suitable for seed germination. The culture was performed in a growth chamber with a 500 μmoles/(m^2^·s) light intensity, 50% relative air humidity, 14 h light/10 h dark, and 25 °C during the day and 20 °C at night. On the 10th whole day after sowing, in each replicate, only one cotton seedling was kept, and the other seedlings were removed. On the 15th whole day after sowing, the same amount of water was added to each replicate to maintain the moisture content. On the 25th whole day, GA_3_, SA, and PP_333_ solutions of different concentrations were applied. Irrigation was performed multiple times with a small amount of water each time to allow the plants to be fully irrigated.

### 4.2. Measurements and Methods

(1)Determination of the root morphological parameters of the seedlings

On the 36th day after sowing, the cotton plants were removed individually from the culture substrate. The entire root system of each plant was removed from the substrate and washed with clean water to remove impurities. Subsequently, the root was placed without overlapping into a transparent glass dish filled with water and scanned as an image file with a root scanner (Epson V800, Seiko Epson Corporation, Nagano Prefecture, Japan), and Win Rhizo software (Regent Instruments, Québec, QC, Canada) was used in the quantitative analysis of the total length, surface area, volume, and diameter.

(2)Determination of the activity of root-related enzymes

After the determination of morphological parameters, the cotton root system of each treatment was frozen in liquid nitrogen for 30 mins, and then stored in a −80 °C freezer before the determination of SP content and SOD and POD activities. The enzyme activity of the root system was determined using the superoxide dismutase assay kit (WST-1 method) to determine SOD activity, peroxidase assay kit (spectrophotometric colorimetry) to determine POD activity, and the total protein assay kit (with standard: BCA method) to determine SP content.

### 4.3. Data Processing and Analysis

Origin 2018 was used to generate figures, and SPSS 25.0 was used for the statistical analysis. Duncan’s new multiple range test (MRT) was used for single factor analysis of variance (ANOVA). The factor analysis in SPSS 25.0 was used for principal component analysis. Root length, root surface area, root volume and root mean diameter were set as morphological factors, soluble protein content was set as the protein content factor, and POD activity and SOD activity were set as enzyme activity factors. The data in the graphs represent the mean ± standard deviation.

## 5. Conclusions

(1)The application of 0.050 mg/L GA_3_ significantly increased the total root length of Z619 and Z27; 0.010 mmol/L SA significantly increased the total root length, total root surface area, and root volume of Z39; and 0.10 mg/L PP_333_ significantly increased the total root length and total root surface area of Z619.(2)The application of 0.80 mg/L GA_3_ increased SOD activity in the roots of Z27 and Z39. All SA concentrations increased SOD activity in the roots of Z619 and Z27. In all varieties, 0.050 mg/L GA_3_ increased the POD activity in the roots, and with the increase of GA_3_ concentration, POD activity showed a decreasing trend. In Z619 and Z39, 0.050 mmol/L SA increased the soluble protein content of the roots. The low PP_333_ concentration increased the POD activity in the Z619 roots, and 0.250 mg/L PP_333_ significantly increased the POD activity in the Z39 roots.(3)Based on the comprehensive analysis of the effects of growth regulators on the root growth and the activities of root enzymes, as well as the results of principal component analysis, 0.10 mmol/L SA was the optimal treatment promoting the root development of Z619, and 0.050 mmol/L SA was the optimal treatment promoting the root development of Z27 and Z39.

## Figures and Tables

**Figure 1 plants-11-02964-f001:**
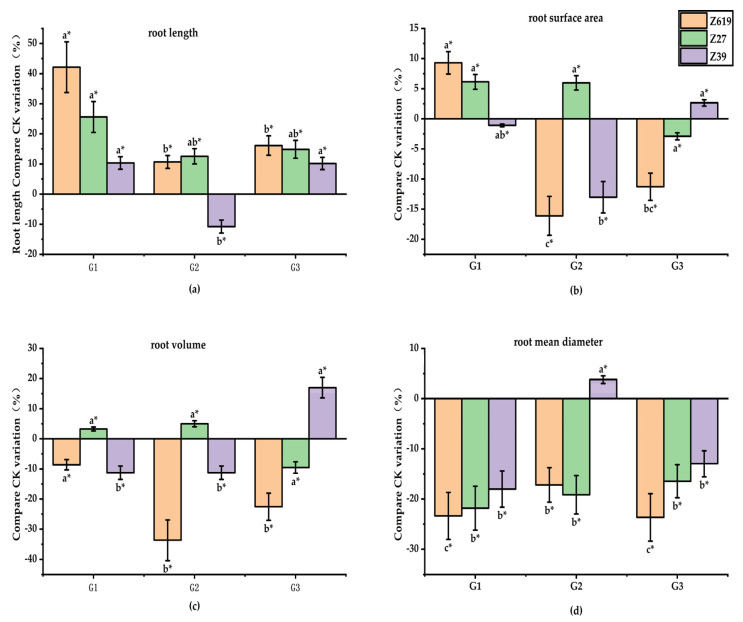
(**a**) Effects of different concentrations of gibberellin on root length; (**b**) Effects of different concentrations of gibberellin on root surface area; (**c**) Effects of different concentrations of gibberellin on root volume; (**d**) Effect of different concentrations of gibberellin on root mean diameter. * Different lowercase letters in the figure indicate that different treatments of the same variety have significant differences at *p* < 0.05.

**Figure 2 plants-11-02964-f002:**
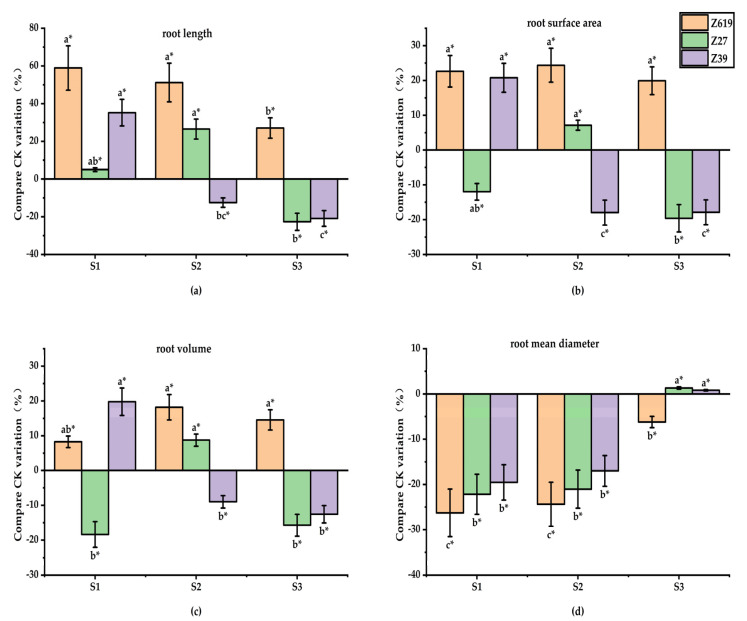
(**a**) Effects of different concentrations of salicylic acid on root length; (**b**) Effects of different concentrations of salicylic acid on root surface area; (**c**) Effects of different concentrations of salicylic acid on root volume; (**d**) Effect of different concentrations of salicylic acid on root mean diameter. * Different lowercase letters in the figure indicate that different treatments of the same variety have significant differences at *p* < 0.05.

**Figure 3 plants-11-02964-f003:**
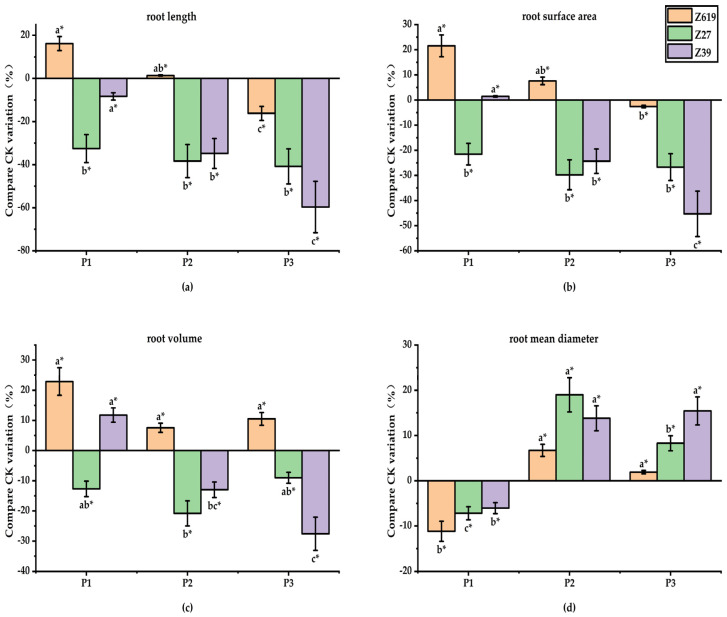
(**a**) Effects of different concentrations of paclobutrazol on root length; (**b**) Effects of different concentrations of paclobutrazol on root surface area; (**c**) Effects of different concentrations of paclobutrazol on root volume; (**d**) Effect of different concentrations of paclobutrazol on root mean diameter. * Different lowercase letters in the figure indicate that different treatments of the same variety have significant differences at *p* < 0.05.

**Figure 4 plants-11-02964-f004:**
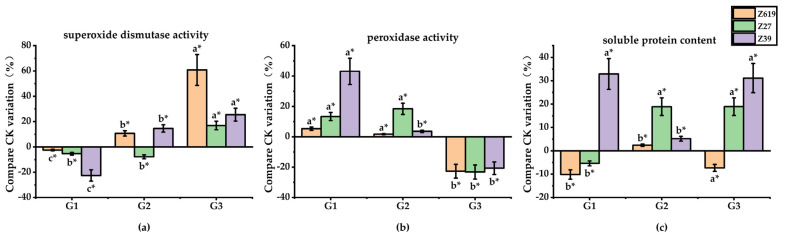
(**a**) Effects of different concentrations of gibberellin on superoxide dismutase activity in root system; (**b**) Effects of different concentrations of gibberellin on peroxidase activity in roots; (**c**) Effect of different concentrations of gibberellin on soluble protein content in roots. * Different lowercase letters in the figure indicate that different treatments of the same variety have significant differences at *p* < 0.05.

**Figure 5 plants-11-02964-f005:**
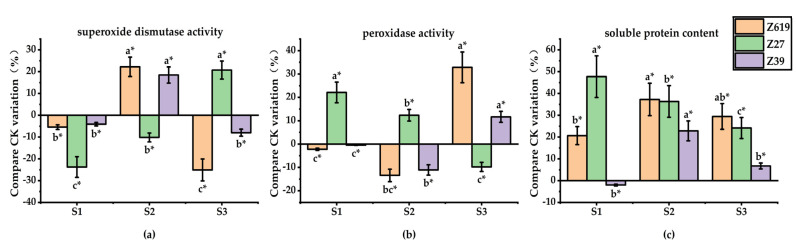
(**a**) Effects of different concentrations of salicylic acid on superoxide dismutase activity in root system; (**b**) Effects of different concentrations of salicylic acid on peroxidase activity in roots; (**c**) Effect of different concentrations of salicylic acid on soluble protein content in roots. * Different lowercase letters in the figure indicate that different treatments of the same variety have significant differences at *p* < 0.05.

**Figure 6 plants-11-02964-f006:**
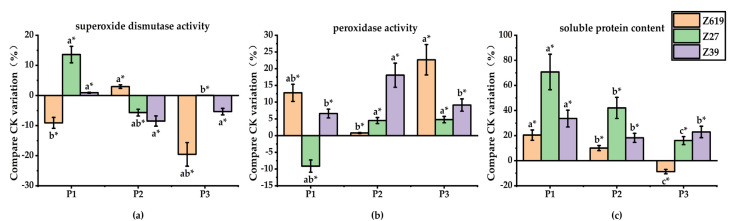
(**a**) Effects of different concentrations of paclobutrazol on superoxide dismutase activity in root system; (**b**) Effects of different concentrations of paclobutrazol on peroxidase activity in roots; (**c**) Effect of different concentrations of paclobutrazol on soluble protein content in roots. * Different lowercase letters in the figure indicate that different treatments of the same variety have significant differences at *p* < 0.05.

**Figure 7 plants-11-02964-f007:**
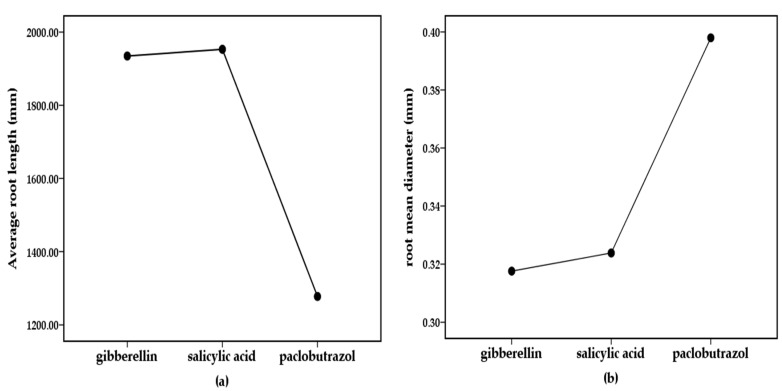
(**a**) Root length with different growth regulators treatments; (**b**) Root mean diameter with different growth regulators treatments.

**Figure 8 plants-11-02964-f008:**
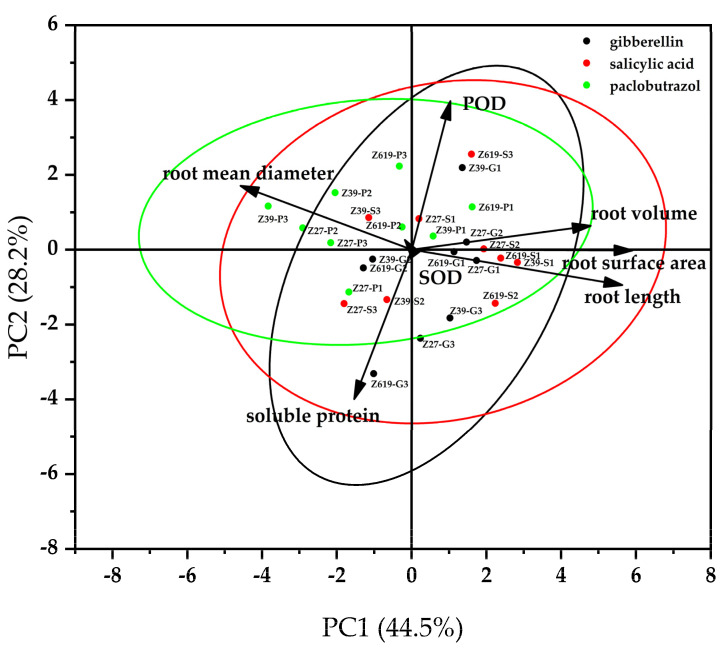
Principal component analysis diagram of cotton variety and growth regulator.

**Table 1 plants-11-02964-t001:** Cotton varieties and growth regulators.

	Cotton Variety	Types of Growth Regulators
1	Zhongmian 619 (Z619)	Gibberellins (GA_3_)
2	Xinluzao 27 (Z27)	Salicylic acid (SA)
3	Xinluzao 39 (Z39)	Paclobutrazol (PP_333_)

**Table 2 plants-11-02964-t002:** Treatments in the experiment.

Cotton Variety	Treatment	Concentration	pH
Z619 Z27 Z39	CK	-	7.0
	G1	0.050 mg/L	6.5
	G2	0.20 mg/L	6.4
	G3	0.80 mg/L	6.2
	S1	0.010 mmol/L	6.6
	S2	0.050 mmol/L	6.6
	S3	0.10 mmol/L	6.3
	P1	0.10 mg/L	6.9
	P2	0.250 mg/L	6.8
	P3	0.50 mg/L	6.8

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
