# Peer review of "Effect of Plant Growth Regulators on Cotton Seedling Root Growth Parameters and Enzyme Activity"

_plants, 2022, doi:10.3390/plants11212964_

Round 1

Reviewer 1 Report

The article "Effect of Plant Growth Regulators on Cotton Seedling Root Growth Parameters and Enzyme Activity" initiated an interesting concept to find suitable plant growth regulators and their optimal concentrations to improve the growth of the cotton seedling root system. However, the manuscript needs to be revised, and some important details need to be addressed. I would like to suggest that you incorporate more details of three cotton varieties and the parameters of the used soil. Include reference/s of the method of activity of root-related enzymes. The results part is well-written and well-justified. Rectify all typing mistakes and grammatical errors. I suggest comparing and discussing your results with the recently published data, so, the discussion needs major revision. Verify that every reference has a corresponding citation within the text and vice versa by cross-referencing all of the citations in the text with the references in the reference section.

Author Response

Response to Reviewer 1 Comments

We would like to thank the Editor and Reviewers for the time and effort in reviewing our manuscript and for providing thoughtful and constructive comments to improve this manuscript. All the comments have been addressed in detail as described below. Note that all line numbers in the response indicate the locations in the revised submission with tracked changes.

The article "Effect of Plant Growth Regulators on Cotton Seedling Root Growth Parameters and Enzyme Activity" initiated an interesting concept to find suitable plant growth regulators and their optimal concentrations to improve the growth of the cotton seedling root system. However, the manuscript needs to be revised, and some important details need to be addressed. I would like to suggest that you incorporate more details of three cotton varieties and the parameters of the used soil. Include reference/s of the method of activity of root-related enzymes. The results part is well-written and well-justified. Rectify all typing mistakes and grammatical errors. I suggest comparing and discussing your results with the recently published data, so, the discussion needs major revision. Verify that every reference has a corresponding citation within the text and vice versa by cross-referencing all of the citations in the text with the references in the reference section.

Point 1: I would like to suggest that you incorporate more details of three cotton varieties and the parameters of the used soil. Include reference/s of the method of activity of root-related enzymes.

Response 1: Thank you very much for your detailed comments, here are our replies: In this study, the nutrient substrate is used to replace the soil to plant cotton. The nutrient substrate is a natural fertilizer formed by decaying leaves and weeds. Therefore, it is rich in organic matter, and its pH is acidic, which is suitable for the growth of most plants. At the beginning of the study, more ideal termination conditions were considered, so there was no index analysis related to soil. As suggested, some references related to root related enzymes have been added (Line 62-74).

Point 2: I suggest comparing and discussing your results with the recently published data, so, the discussion needs major revision. Verify that every reference has a corresponding citation within the text and vice versa by cross-referencing all of the citations in the text with the references in the reference section.

Response 2: Thank you for the suggestions. We have compared the results with the recently published data, and focused on revising the discussion part. (Line 366-390).

Reviewer 2 Report

The work entitled “Effect of Plant Growth Regulators on Cotton Seedling Root Growth Parameters and Enzyme Activity” fits with the aim of the journal Plants MDPI. The authors investigated the effect of growth regulators on cotton seedling root parameters and enzyme activity. The manuscript is fascinating as it deals with a concrete problem of the Tarim Basin and tries to find a solution.

Abstract

The first part of the abstract (lines 10-15) is not clear, please rewrite it.

Introduction

56-68 references need to be cited, much information is not referenced.

Results

In general, this section is not clear. Please rewrite the Results section in a more simple way.

Line 94: “Bulleted lists look like this:” I think that this is a typo, please delete it.

In all figures, please include the level of significance (*, **, ***, or NS) for each treatment. Furthermore, in my opinion, it is better to include the parameter name also in the figure itself and not only in the caption for better viewing.

Since this journal's Results section is presented before the Material and Methods section, treatment abbreviations should be explained (e.g. CK: control, etc.).

Paragraphs 2.3 and 2.4 must be included in the Materials and Methods section. In the Results section, the authors must discuss the statistical analysis outcomes when they describe the data. Moreover, in my opinion, Tables 2 and 3 should be included in the Supplementary materials section to lighten the manuscript. Whereas, in my opinion, table 1 must be deleted and the information included in the figures. When the interaction between the two treatments is not significant, a graphic of the single effects of the treatments (varieties and growth regulators) is required.

A PCA graphic is suggested to better visualize the PCA output, it must include all treatments and all variables.

In figures 4, 5, and 6 the letters that indicate the mean separation are missing, please provide them.

Discussion

 Lines 379-383 this is a conclusion phrase, not a discussion.

Materials and methods

Protocol used to determine the activity of root-related enzymes must be included. In paragraph 4.3 authors report “The data in the graphs represent the mean ± standard deviation”, however, I did not find the standard deviation bars, please include standard deviation or standard error in each graph.

References

The reference section has formatting issues, please correct them.

Final remarks

The manuscript has some flaws that need to be addressed, thus I recommend accepting the paper only after the major revisions I suggest.

Author Response

Response to Reviewer 2 Comments

We would like to thank the Editor and Reviewers for the time and effort in reviewing our manuscript and for providing thoughtful and constructive comments to improve this manuscript. All the comments have been addressed in detail as described below. Note that all line numbers in the response indicate the locations in the revised submission with tracked changes.

The work entitled Effect of Plant Growth Regulators on Cotton Seedling Root Growth Parameters and Enzyme Activity" fits with the aim of the journal Plants MDPI. The authors investigated the effect of growth regulators on cotton seedling root parameters and enzyme activity. The manuscript is fascinating as it deals with a concrete problem of the Tarim Basin and tries to find a solution.

Thank you very much for your detailed comments, here are our replies.

Point 1: Abstract

The first part of the abstract (lines 10-15) is not clear, please rewrite it.

Response 1: Thank you for the suggestion. The first part of the abstract were rewritten as follows. (Line 10-16).

It is well known that the survival rate of cotton seedlings is low, and the growth and development status at this stage is crucial to improve productivity. Plant hormones are important factors to promote the growth and development of cotton seedlings. Growth regulators have the same function as plant hormones. The purpose of this research is to explore the effects of different concentrations of growth regulators on cotton root morphological parameters and enzyme activities, and to find the suitable plant growth regulators and their optimal concentrations to improve the growth of the cotton seedling root system.

Point 2: Introduction

56-68 references need to be cited, much information is not referenced.

Response 2: Thank you for the suggestion. We have modified the introduction and added more references. (Line 62-72).

Point 3: Results

In general, this section is not clear. Please rewrite the Results section in a more simple way.

Line 94: "Bulleted lists look like this." I think that this is a typo, please delete it.

Response: Thank you for the suggestion. We have deleted it. (Line 98).

In all figures, please include the level of significance (*, **, *** or NS) for each treatment. Furthermore, in my opinion, it is better to include the parameter name also in the figure itself and not only in the caption for better viewing.

Response: Thank you for the suggestions. We have added the level of significance (*) and parameter name of each treatment in all figures. (Line 124, 157, 190, 228, 259, 291).

Since this journal's Results section is presented before the Material and Methods section, treatment abbreviations should be explained (e.g. CK: control, etc.).

Response: Thank you for the suggestion. We explained the treatment abbreviations in the results section. (Line 100-103, 138-145, 169-172).

Paragraphs 2.3 and 2.4 must be included in the Materials and Methods section. In the Results section, the authors must discuss the statistical analysis outcomes when they describe the data. Moreover, in my opinion, Tables 2 and 3 should be included in the Supplementary materials section to lighten the manuscript. Whereas, in my opinion, table 1 must be deleted and the information included in the figures. When the interaction between the two treatments is not significant, a graphic of the single effects of the treatments (varieties and growth regulators) is required.

Response: Thank you for the suggestions. We have added the analysis methods in paragraphs 2.3 and 2.4 to the Materials and Methods section. (Line 467-472). And we deleted Table 1 because we added a single factor analysis graphic.

A PCA graphic is suggested to better visualize the PCA output, it must include all treatments and all variables.

Response: Thank you for the suggestion. We drew the PCA graphic. (Line 254).

In figures 4, 5, and 6 the letters that indicate the mean separation are missing, please provide them.

Response: Thank you for the suggestion. We added the letters representing the mean separation in figures 4, 5, and 6. (Line 228, 259, 291).

Point 4: Discussion

Lines 379-383 this is a conclusion phrase, not a discussion.

Response 4: Thank you for the suggestion. We deleted 379-383 lines and modified them to put them in other places. (Line 383-388).

Point 5: Materials and methods

Protocol used to determine the activity of root-related enzymes must be included. In paragraph 4.3 authors report "The data in the graphs represent the mean ± standard deviation", however, I did not find the standard deviation bars, please include standard deviation or standard error in each graph.

Response 5: Thank you for the suggestions. We have added the Protocol used to determine the activity of root-related enzymes to the Materials and Methods section. (Line 459-464). We have added standard deviation bars to all figures. (Line 62-74).

Point 6: 6.References

The reference section has formatting issues, please correct them.

Response 6: Thank you for the suggestion. We have revised the formatting issues of the reference section. (Line 496-560).

Reviewer 3 Report

This study is rudimentary in design and presentation. I believe it's a local extension report, not an original research paper. 

The manuscript would not be interesting for the universal audience. 

The traits are not enough to decide on the reproducibility of the research.

There is no logical crosstalk between the root traits and the enzymes which have been measured.

Author Response

Response to Reviewer 3 Comments

We would like to thank the Editor and Reviewers for the time and effort in reviewing our manuscript and for providing thoughtful and constructive comments to improve this manuscript. All the comments have been addressed in detail as described below. Note that all line numbers in the response indicate the locations in the revised submission with tracked changes.

This study is rudimentary in design and presentation. I believe it's a local extension report, not an original research paper.

The manuscript would not be interesting for the universal audience.

The traits are not enough to decide on the reproducibility of the research.

There is no logical crosstalk between the root traits and the enzymes which have been measured.

Response : Thank you for taking the time to read this article and your valuable suggestions. As this study is located in the Tarim Basin, Xinjiang, which is an arid area, the soil salinization is serious and the environment is harsh. In addition, local agricultural technology is underdeveloped, and local farmers need a convenient and inexpensive way to improve cotton growth and yield. After the completion of this study, we will study the effects of growth regulators on the growth and development of cotton under drought and saline alkali stress, hoping to be of great help to the local agricultural and economic development.

Round 2

Reviewer 2 Report

The authors addressed all points of criticism, thus the manuscript can be accepted in present form.

Reviewer 3 Report

I still do believe that the manuscript is not enough informative to attract the universal audiences and even after some corrections, it is not worthy of acceptance with your journal. But to be fair, please try another expert reviewers ideas to reach a solid and reliable decision.